# An Ontological Model for the Representation of Vallenato as Cultural Heritage in a Context-Aware System

**María Antonia Diaz-Mendoza** [1,*], **Emiro De-La-Hoz-Franco** [1], **Jorge Eliecer Gómez Gómez** [2] **and Raúl Ramírez-Velarde** [3,*]

1   Department of Computer Science and Electronics, Universidad de la Costa CUC, Barranquilla 080002, Colombia; edelahoz@cuc.edu.co
2   Department of System, Universidad de Córdoba, Montería 230002, Colombia; jeliecergomez@correo.unicordoba.edu.co
3   Tecnológico de Monterrey, Monterrey 64849, Mexico
*   Correspondence: mdiaz60@cuc.edu.co (M.A.D.-M.); rramirez@tec.mx (R.R.-V.); Tel.: +57-31-6576-4745 (M.A.D.-M.)

**Abstract:** The traditional Colombian vallenato was declared Intangible Cultural Heritage of Humanity by UNESCO on 1 December 2015 with urgency for it to be safeguarded, which led the government of Colombia in the head of the Ministry of Culture and the vallenato music cluster to develop a safeguarding plan that contains different activities, among which stands out a platform for the management of vallenato through educational processes. In this sense, this document proposes an ontological model for the representation of vallenato as cultural heritage in a context-aware system called Vallenatic. The ontology was developed using the NeOn methodology, designed in the Protégé software, and has 15 concepts (Vallenata Songs, Artist, Devices, Persons, Environment, Cultural Managers, Interface, Location, MOOC, Learning Object, Profile, Preference, Net, Time Cultural, Event, Cultural_Sites). The model was evaluated by means of nine (9) situations described in natural language and SWRL; this language was used since it allows expressing OWL concepts in combination with RuleML. The proposed model can be used for other musical genres that have the recognition of cultural and intangible heritage, such as the Spanish flamenco, Argentine Tango, Mexican Mariachi music, Peruvian scissors dance, Brazilian capoeira, Dominican bachata, Jamaican reggae, among others.

**Keywords:** ontological model; context-aware; vallenato; cultural heritage; semantic web



## 1. Introduction

In the Republic of Colombia, the Caribbean region, made up mainly of the departments of Cesar, La Guajira, and Magdalena, has a cultural richness that characterizes it, its gastronomy [1], their different oral traditions [2], and its musical genres, and it gives this area of the country a distinction that identifies it both nationally and internationally. Part of this seal is configured by the so-called vallenata culture, a symbiosis of traditions of the indigenous people, black Africans, and Spaniards that converged in the land of Magdalena Grande and who were accompanied with instruments, narrating their daily experiences and giving birth to the traditional vallenata music, which has been considered a reference in the forging of the cultural identity of the Colombian Caribbean [3] to such an extent that the Nobel Prize winner Gabriel García Márquez says that it was precisely the artistic and literary richness of the vallenato songs he heard in his native Aracataca that inspired him to tell stories through his writings.

This has been the representative music of Colombia, which is defined by Consuelo Araujo Noguera [4] as "the chronicle, made music, of a whole people. They are events narrated in the compositions, which may well have been performed by a specific person, but which are in themselves the reflection of a thousand other events, identically the same, which occurred to so many other people".

Despite all the cultural richness that endows the vallenato, its folkloric and literary essence has been threatened by the fusions to which it has been exposed, to the commercialization of its lyrics that threatens the costumeries chronicle that characterizes this musical genre, for which measures must be taken to preserve its tradition, such as the one adopted by the Government of the Department of Cesar and the Ministry of Culture of Colombia, which, on 1 December 2015, led UNESCO to declare vallenato as "Cultural and Intangible Heritage of Humanity" that urgently needs to be safeguarded [5].

Different vallenato composers, such as Rosendo Romero [6], establish that one of the best ways to safeguard vallenato is through education. It is important to disseminate in the new generations the cultural heritage inherent to vallenato, executing different strategies in schools, such as teaching the use of the accordion, promoting the learning of the traditional roots of the Colombian Caribbean region, providing spaces where the poetic content of classic vallenato songs is analyzed and understood, teaching the differentiation of the musical genres of vallenato (puya, paseo, merengue, and son), in which they build their own lyrics and validate the compositions from the approach of the chronicle and narrative of this type of music.

Romero's proposal was accepted by the government of the department of Cesar and the municipality of Valledupar—Colombia, (considered the world capital of vallenato), where education and the use of the different means of dissemination offered by technology are established as one of the possible ways to safeguard traditional vallenato music by applying current strategies that are attractive to children, adolescents, and young people, thus ensuring that the customs and traditions surrounding this musical genre are preserved and disseminated over the years.

Various technological tools are used for the protection of cultural heritage worldwide; UNESCO itself on its website has an inventory of all the assets that enjoy the distinction of intangible cultural heritage, allowing its consultation and the dissemination of the various activities aimed at safeguarding it.

When reviewing the inventory, it is observed that the Colombian vallenato does not currently have any technological tool designed for this purpose; even though the message given by UNESCO is of "urgency", technology has not yet been used to safeguard the representative music of the northern coast of Colombia.

This paper presents a proposal for a MOOC-based contextual awareness ontology for the management of traditional Colombian vallenato. A context-based ontological model for the preservation of vallenato as cultural heritage is one that allows the interaction of users with elements of the context such as: the devices, the composers, the users, the physical environment, the different cultural managers, the instructors or historians, the interface, the location, the MOOCs as a strategy for the teaching of vallenato and the artists.

biomolecules-2457539

This proposed model can be adapted to other musical genres that also enjoy recognition as Intangible Cultural Heritage of Humanity status, such as the Spanish flamenco, Argentine Tango, Mexican Mariachi music, Peruvian scissors dance, Brazilian capoeira, Dominican bachata, Jamaican reggae, among others.

## 2. Related Work

For this section, a systematic literature review was conducted on ontology designs for cultural heritage, where an exhaustive search was conducted in the databases Scopus, Web of Science, IEE Xplore, and Science Direct, using the following keywords: "Cultural Heritage", "Ontology"; "Technology", "context aware"; resulting in the following search strings: "Ontological Model AND Cultural Heritage"; "context aware AND Cultural Heritage "; "Ontology AND Cultural Heritage". In addition, the articles that were Open Access between the years 2028 and 2022 were selected. The review resulted in 259 articles, of which 54 were in more than one database (duplicates), finally analyzing 204 articles in total.

As a result of the SLR, it was found that there are different ontological models related to cultural heritage as well as related to MOOC type courses and educational environments,

but no model was found that related these two components. The most relevant studies resulting from the SLR are listed below.

Much of the literature that shows that different ontological models have been developed for CH, the best known being the CIDOC CRM (https:www.CIDOC-CRM.org/, accessed on 3 February 2023), conceptual reference model. A widely developed ontology that provides a semantic framework for the integration, mediation, and access to CH information, it can handle different types of information, such as GLAMs (this is the name given to articles published by Galleries, Libraries, Archives, and Museums.), as it is the only conceptual model so far that corresponds to the ISO standard on CH, specifically ISO 21127 of 2006. It has 99 classes and 188 properties [7].

As mentioned above, the CDOC CRM ontology is the most widely used ontology in CH and has been adapted in different projects, such as the proposed in [8], where the authors conducted a study whose objective is to develop an ontological model for heritage sites in Iran, specifically the Sa'dabad complex, using concepts from the main ontology developed for cultural heritage called CIDOC CRM integrated with GeoSPARQL, the standard ontology in the geospatial field, to incorporate spatial semantics with heritage information. This integration is for the purpose of allowing users to freely explore any information they want about the heritage site of their choice; CIDOC CRM provides the heritage information, and GeoSPARQL provides the location of the site. Ontology was developed in Protégé, composed of 2 superclasses: Entity by CIDOC CRM with 15 subclasses and Spatial Object by GeoSPARQL with 5 subclasses. It has 25 concepts with a hierarchy of five levels of depth.

As there are different ontologies designed to protect the heritage of different countries, in [9] how a Bulgarian ontology network entitled Cultural and Historical Heritage Ontology Network (CHH-OntoNet), where six ontologies developed in the Protégé application are structured, hierarchically organized and separated from the architecture as a separate module. Among the ontologies present in this project, the following stands out: the Agents ontology includes 1302 axioms, 61 classes, 54 object properties, 8 data properties, and 124 individuals, and it is designed to protect Bulgaria's architectural heritage [10].

Similarly, in [11], a network of ontologies is presented to represent cultural heritage data and to publish the General Catalog proposed by the Italian Ministry of Culture. This ontology network is designed for cultural heritage management, taking classes from the following ontologies: CIDOC CRM, EDM, ORE, FOAF, DC, and SKOS.

In [12], an ontology is presented for the representation of cultural heritage considering the UNESCO definitions; this ontological model is called CURIOCITY (Cultural heritage for urban tourism in indoor and outdoor environments of the CITY). CURIOCITY ontology has a three-level architecture (Upper, Middle, and Lower ontologies) in accordance with a purpose of modularity and levels of specificity.

In [13], an ontology designed for the purpose of solving semantic heterogeneity among museum data sources is presented. For this purpose, a high-level ontology architecture is used and a high-level ontology for museums is built to solve the problem of semantic heterogeneity among museum data sources. This ontology is tested by means of a case study where a local ontology is built to manage heritage information related to New Zealand soldiers who participated in World War II. This ontology for cultural heritage management takes classes from the following ontologies: CIDOC CRM, EDM, ORE, FOAF, DC, and SKOS and is called M.O.M. (Museum Ontology-Based Metadata).

Considering that the meaning of this ontological model complements the management of cultural heritage with formative processes, works related to knowledge management through ontological models are presented, such as the one presented in [14] where they propose a system based on environmental information to support active learning, which takes into account the context and its reasoning, which is integrated into the system architecture. Context reasoning is used to efficiently provide learning resources to the student based on contextual information, such as the location of the individual, the time and date of the activity, the interaction with objects, and the profile. The system is validated in

three different programs, such as medicine, nursing, and systems engineering, with control and experimental groups. The results obtained from the experimental validation of the contextual awareness system as a support for active learning showed that the experimental groups of the three programs obtained a higher average academic performance than the control groups during the post-test. Similarly, the hypothesis tests of the experiments showed that there were significant differences in academic performance in favor of the experimental groups in question.

In [15], an ontology-based framework for an adaptive learning system for an adaptive learning system detailed information contextual categorization and modeling along with the use of ontology to explicitly specify and modeling along with the use of the ontology to explicitly specify the learner's learner context in an e-learning environment. This ontology-based context model contains semantic relationships between concepts and can provide semantic-based context information for searching learning material in context-based e-learning environments.

In [16], an intelligent learning environment (SLE) developed in a simple form is shown to be easy to understand and use to develop learning environments that support effective, efficient, and engaging learning processes. The proposed model adopted the architecture of the intelligent tutoring system as its basis. The model components were generated by mapping 12 existing SLE models, frameworks, and best practices. Model training and validation followed the system dynamics (SD) modeling process. In addition, the indirect model validation process was also conducted by interviewing educational technology practitioners and experts based on the evaluation results of two recently developed SLE instances. To measure the maturity of existing SLE instances, this study also derived the SLE maturity model. This work establishes that the proposed model is easy to use and understand and easy to evaluate.

In [17], an EduAdapt model is proposed: an architectural model for the adaptation of learning objects considering the characteristics of the device, the learning style, and other information of the student's context. In this sense, this proposal uses ingests and proposes an ontological model, which they called OntoAdapt. The authors consider that ontology can help to recommend learning objects to learners or adapt these objects according to the context (context-aware computing). The proposal was evaluated using scenarios and metrics to evaluate the ontology as well as by developing a prototype of the EduAdapt model, which was submitted to one of 20 students with the intention of evaluating the usability and adherence to the adapted objects, resulting in a 78% acceptance rate.

## 3. Ontology Model

### 3.1. Selected Methodology

One of the main steps to model ontologies was to select an appropriate methodology; in this sense, there were many proposals as shown in Table 1.

**Table 1.** Methodologies for modeling web ontologies.

| Uschold y King (1995) [18] | Grüninger and Fox (1995)—TOVE Methodology [19] | Kactus (1997) [20,21] | Fernández-López, Gómez-Pérez and Juristo (1997)—Methontology [22,23] | On-To-Knowledge (2001) [24] | Ontology Development 101 | Noy y McGuinness (2001)—Simple Knowledge-Engineering Methodology | NeOn Methodology (2010) | Stuart (2016) |
|---|---|---|---|---|---|---|---|---|
| 1. Identify the purpose | 1. Determine the competence of the ontology | 1. Application Specification | 1. Specification | 1. Feasibility study | 1. Determine the domain and scope of the ontology | 1. Determine the domain and scope of the ontology | Scenario 1: Ontology networks from the application specification. | 1. Scope of the ontology |
| 2. Construct the ontology a. Capture the ontology b. Encode the ontology c. Integrate with existing ontologies | 2. Define ontology terminology | 2. Preliminary design based on relevant toplevel ontological categories | 2. Knowledge acquisition | 2. Launch. (Delimitation of domains, statement of objectives, extraction of information sources, description of users and future applications). | 2. Determine the intent to use the ontology | 2. Consider reusing existing ontologies | Scenario 2: The reuse and reengineering of non-ontological resources (NOR). | 2. Reuse of the ontology |
| 3. Evaluate | 3. Specify terminology definitions and restrictions | 3. Refinement and structuring of ontology. | 3. Conceptualization | 3. Improvement | 3. Reuse existing controlled ontologies or vocabularies. | 3. List important terms in the ontology | Scenario 3: Development of ontology networks through reuse of ontological resources. | 3. Identifying the appropriate software |
| 4. Document | 4. Test the competence of ontology to demonstrate the integrity of theories | | 4. Integration | 3. Evaluation | 4. List the important domain terms. | 4. Define classes and hierarchies | Scenario 4. Development of ontology networks through reuse and reengineering of ontological resources. | 4. Knowledge acquisition |
| | | | 5. Implementation | 4. Maintenance | 5. Define class hierarchy. | 5. Define class properties | Scenario 5. Development of ontology networks through reuse and mixing of ontological resources. | 5. Identification of important terms |
| | | | 6. Evaluation | | 6. Create the instances. | 6. Define the facets | Scenario 6. Development of ontology networks through reuse, mixing and reengineering of ontological resources. | 6. Identification of additional terms, attributes, and relationships |
| | | | 7. Documentation | | | 7. Create instances | Scenario 7. Development of ontology networks through reuse of ontological design patterns (ODPs). | 7. Specification of definitions |
| | | | | | | | Scenario 8. Development of ontology networks through restructuring of ontological resources. | 8. Integration with existing ontologies |
| | | | | | | | Scenario 9. Development of ontology networks through the localization of ontological resources. | 9. Implementation |
| | | | | | | | | 10. Evaluation |
| | | | | | | | | 11. Documentation |
| | | | | | | | | 12. Sustainability |

To model the proposed ontology, the NeOn methodology [25] applied in [26] was selected. Unlike the other ontology construction models, the NeOn methodology arises from the need to fill the gaps that could not be filled by the three most known models up to that moment, such as the concept of network de ontology and the dimension of collaboration, context, and dynamism [26].

This methodology is based on addressing different scenarios or paths for the construction of ontology and ontology networks. These scenarios are flexible and can be combined with each other, which makes it a methodology that adapts to the needs and different requirements of specific users.

The scenarios are as follows:

- Scenario 1. Development of ontology networks from specification to implementation;
- Scenario 2. Development of ontology networks through reuse and reengineering of non-ontology resources;

- Scenario 3. Development of ontology networks by reusing ontology resources;
- Scenario 4. Development of ontology networks through reuse and reengineering of ontology resources;
- Scenario 5. Development of ontology networks by reusing and mixing ontology resources;
- Scenario 6. Development of ontology networks through reuse, mixing and reengineering of ontology resources;
- Scenario 7. Development of ontology networks by reusing ontology design patterns;
- Scenario 8. Development of ontology networks by restructuring ontology resources;
- Scenario 9. Development of ontology networks by localization of ontology resources.

In this case, scenario 1 was taken, following the methodology established by [26] where it was recommended to follow the tasks presented in Table 1.

### 3.2. Specification of Requirements

The NeOn methodology established an order for the specification of requirements in addition to providing a methodological guide to achieve this activity. The guide established specific tasks summarized by [26] summarizes as shown in Table 2:

**Table 2.** Tasks for requirements specification according to NeOn methodology based on [25].

| Task | What They Do | How They Do It | Who Does It |
|------|-------------|----------------|-------------|
| 1 | Identification of the purpose, scope, and implementation language of the ontology | A series of interviews with users and domain experts should be conducted. | Ontology Development Team |
| 2 | Identification of the intended end users. | A series of interviews with users and domain experts should be conducted. | Ontology Development Team |
| 3 | Identification of intended uses | A series of interviews with users and domain experts should be conducted. | Ontology Development Team |
| 4 | Identification of requirements. | A series of interviews with users and domain experts should be conducted. | Ontology Development Team, Experts. |
|  | The requirements are divided into two types: |  |  |
|  | Non-functional requirements that refer to general aspects not related to the knowledge to be represented by the ontology. |  |  |
|  | Functional requirements that are content requirements related to the knowledge that the ontology should represent. These requirements should be written in the form of competency questions with their respective answers. |  |  |
| 5 | Grouping of the functional requirements. | Exhaustive analysis of what was obtained in the previous task. | Ontology Development Team |

Task 1: Identification of the purpose, scope, and implementation language of the ontology:

- Purpose

The objective of the ontology network was to represent the concepts and the types of contexts that influenced the concepts to adapt their behavior to an adaptive application to promote the vallenata cultural identity with respect to the situation that the user activates.

- Scope

  The Ontology comprised the following concepts:

  ○      Artists (Composers, Musicians);
  ○      Cultural Event;
  ○      Cultural Managers (Managers, Historians, Researchers, Event organizers);
  ○      Cultural_Site;
  ○      Devices;
  ○      Environment;
  ○      Interface;
  ○      Learning Object;
  ○      Localization;
  ○      MOOC;
  ○      Persons (Instructors, Student, Tourist);
  ○      Preference;
  ○      Profile;
  ○      Red;
  ○      Time;
  ○      Vallenata Songs.

- Implementation language

  The ontology was modeled using the Protégé Software in the OWL language.

Task 2: Identification of the intended end users.

During the initial phase of the project, the following intended users were established (see Table 3):

**Table 3.** Expected Users.

| | |
|---|---|
| General users | Students |
| | Tourists |
| | People interested in learning about Vallenato |
| | Arts teachers |
| | Physical Education Teachers |
| | Visitors to the Vallenato Music Cultural Center |
| | Academic |
| Artists related to vallenato music | Singers |
| | Singer |
| | Composers |
| | Accordionists |
| | Guacharaqueros |
| | Cashiers |
| | Chorus |
| | Arrangers |
| | Congas |
| | Drummer |
| | Pianist |
| | Sound Engineer |
| | Short |
| | Euphonium |
| | Clarinet |
| | Trombone |
| | Guitarist |
| | Representatives |
| Cultural managers related to Vallenato | Representatives of Vallenato Music Schools |
| | Representatives of Foundations related to Vallenato |
| | Vallenato Historians |
| | Members of Vallenato festival boards |
| | Teachers |
| | Cultural managers |
| | Rectors of Educational Institutions |
| | Government Institutions |

Task 3: Identification of intended uses

- Store and edit information about users (Students, tourists, musicians, Instructors, other users);
- Store and edit information about the context of users;
- Store and edit information about MOOCs;
- Store and edit information about the devices that users can access;
- Store and edit users' network information;
- Store and edit location information;
- Store and edit information about weather;
- Store and edit information about the users' environment (Artists, Persons, Cultural Managers);
- Store and edit user interface information;
- Store and edit information about vallenato related cultural events;
- Store and edit information about user preferences, profiles, and user roles;
- Store and edit information about vallenato songs.

Task 4: Identification of requirements.

- Non-functional requirements

For the selection of the non-functional requirements, a review of different ontologies with similar purposes that were already designed was established; thus, to establish which requirements of this type were the most used in this sense, the non-functional requirements were the following:

1. The ontology network must be Modular;
2. The ontology network must be in English.

- Functional requirements

To determine the functional requirements for applying the NeOn methodology, competency questions were asked, and five (5) experts were used to divide the sub-concepts of the ontology.

Task 5: Grouping of the functional requirements

In this case, since the functional requirements were defined by concepts and sub-concepts, it was not necessary to group them.

Task 6: Validation of the set of requirements

Both functional and non-functional requirements were reviewed and validated by the ontology development group.

Task 7: Prioritization of the set of requirements

For the non-functional requirements, it was decided to give the same level of priority to both requirements:
Priority 1:

1. The ontology network must be Modular;
2. The ontology network must be in English.

The following level of priority was established for the functional requirements:
Priority 1:
Environment, Location, Preferences/Profile/Role, vallenata songs, Cultural Event, Artist, Cultural Managers, Persons, MOOC, Cultural Site, and Learning Object;
Priority 2:
Time;
Priority 3:
Device;
Priority 4:
Interface and Network.

Task 8: Extraction of terminology and its frequency.

The extraction of terms and their frequency was done using a syntactic annotator, and then, the terms were extracted in a Word document and organized in a graphical presentation (Word Cloud) using an online tool (https://classic.wordclouds.com/). As an example, the word cloud of the concept of vallenato songs were added.

As can be seen in Figure 1, the words with the highest frequencies were those related to the genres of vallenato songs (merengue, son, paseo, and puya) followed by singers, inspiration, lyrics, nature, among others.

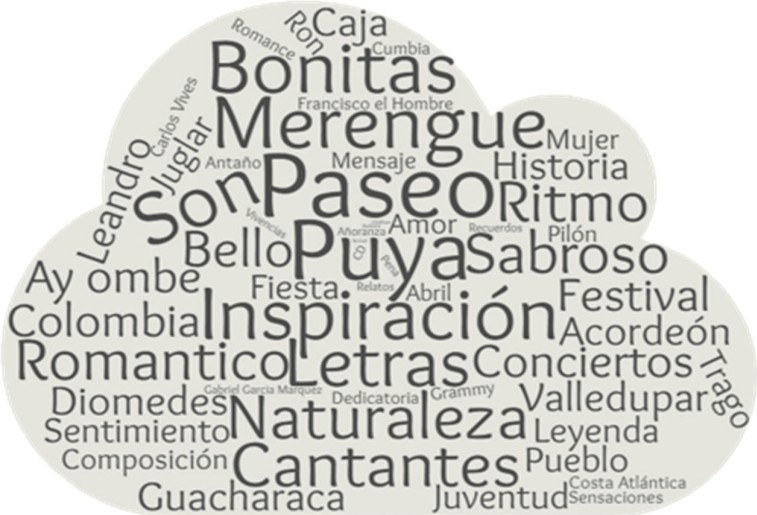

**Figure 1.** Word Cloud Vallenato Songs.

This terminology was useful to get an idea of what the most important terms of the ontology being developed were as well as to obtain a set of terms from which to search for resources, both ontological and non-ontological, that could be reused during ontology development [26].

*3.3. Graphic Design of the Ontology*

The ontology designed had 15 concepts and their respective relationships, which can be seen in Figure 2.

The ontological network of the proposed system was constituted by classes that identify the dimensions of the context for the management of vallenato as national heritage. The following is a brief description:

- **Vallenatas Songs:** Describes the information of the vallenato compositions, such as lyrics, genre (merengue, paseo, puya, and son), duration;
- **Artists:** Describes the information of the composers and musicians related to vallenato, such as their personal information and relates the songs composed (composers) and performed (musicians);
- **Devices:** Describes the devices, both software and hardware, in context;
- **Environment:** Describes information about the environment of people, artists, and cultural managers;
- **People:** Describes the information of the people involved in the context, such as students (those enrolled in the MOOCs), instructors (those who teach using the platform, those who make use of the MOOCs), and tourists (describes the visitors from anywhere in the world who attend the vallenato events);
- **Cultural Event:** Describes the information of the events related to the vallenato, these events can be festivals, concerts, talks, private parties, among others;

- **Cultural Managers:** Describes the information of the people who promote Colombian vallenato from different fronts (event organizers, artists' representatives, historians, and researchers);
- **Interface:** Describes the interface of the MOOCs and the learning objects inspired by the vallenato culture;
- **Location:** Describes the location of the context of all the elements that make up the model;
- **MOOC:** Describes the information of the massive open online courses to teach about vallenata culture that can be developed in the context;
- **Learning Object:** Describes the different learning objects based on vallenata culture used in the context;
- **Profile:** Describes the profile of the actors (people, artists, cultural managers) involved in the process;
- **Preference:** Describes the preferences of the actors (people, artists, cultural managers) involved in the process;
- **Network:** Describes the characteristics of the network connectivity of the devices owned by the actors (people, artists, cultural managers) involved in the process;
- **Cultural sites:** Describes the cultural sites related to vallenato;
- **Time:** Describes the notion of time in the context of vallenato.

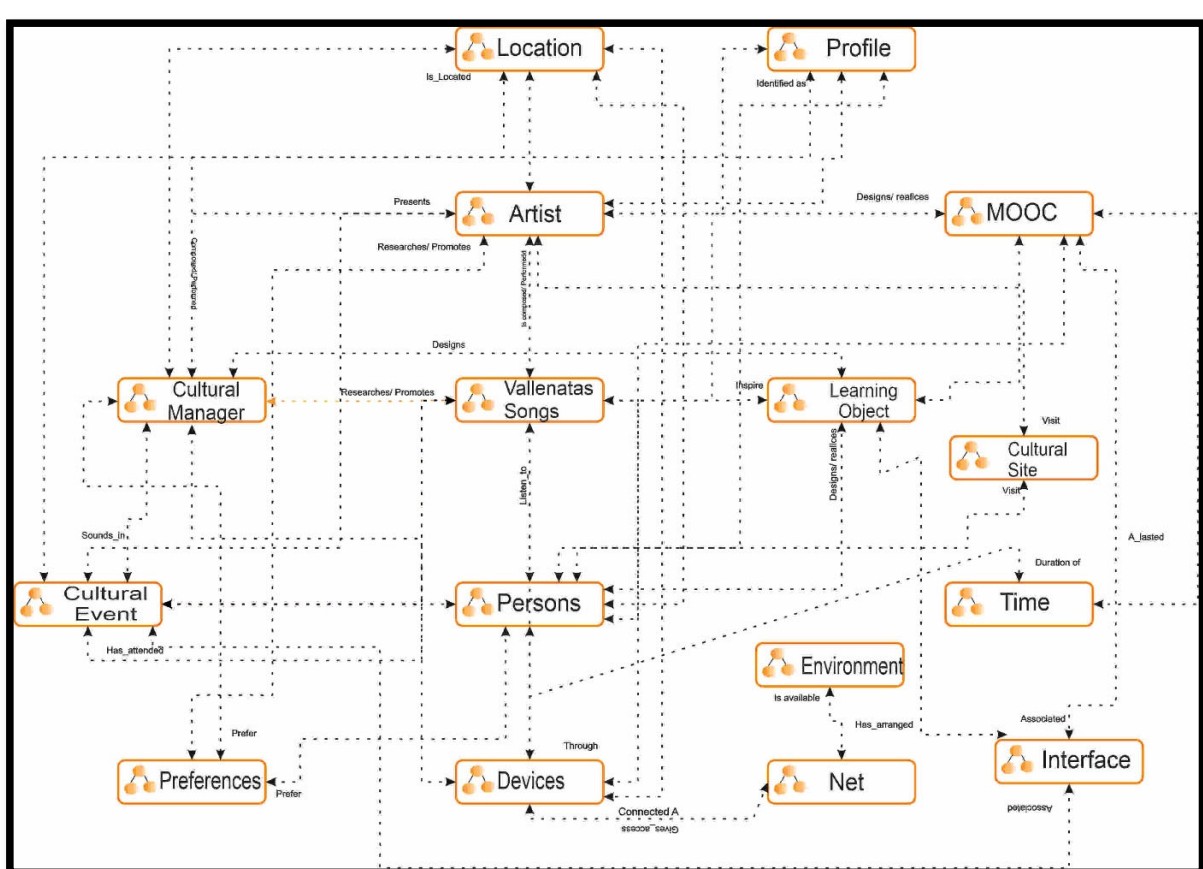

**Figure 2.** Ontological Model.

### 4. Ontology in Protégé

As specified above, for the ontology modeling, we used the application Protégé (https://protege.stanford.edu/, accessed on 3 February 2023), developed at Stanford University. Protégé is a free open-source platform that provides a set of tools for building domain models and knowledge-based applications with ontologies, which has been widely used in different projects [8,12,27–29].

The version of Protégé used was 5.6.1, downloaded directly from the site https://protege.stanford.edu/products.phpthe (accessed on 3 February 2023). The downloadable version was chosen because it allows for designing and/or modifying more than one ontology using the same interface that can be customized. In addition, it has very good support.

After the application configuration process, the classes were created as shown in Figure 3.

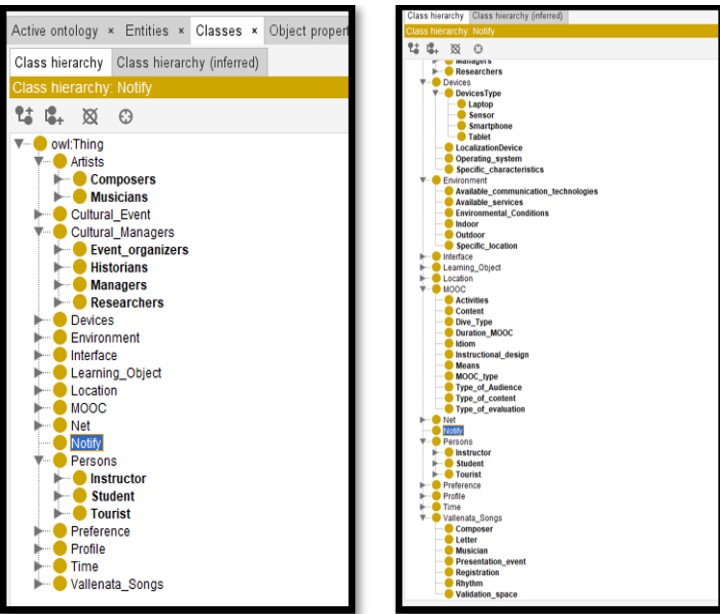

**Figure 3.** Class and subclass. Obtained from Protégé software, version 5.6.1.

Once the 15 classes that made up the ontology and their respective subclasses were created, we proceeded to create the object properties, data properties, and individuals by class. See Figure 4.

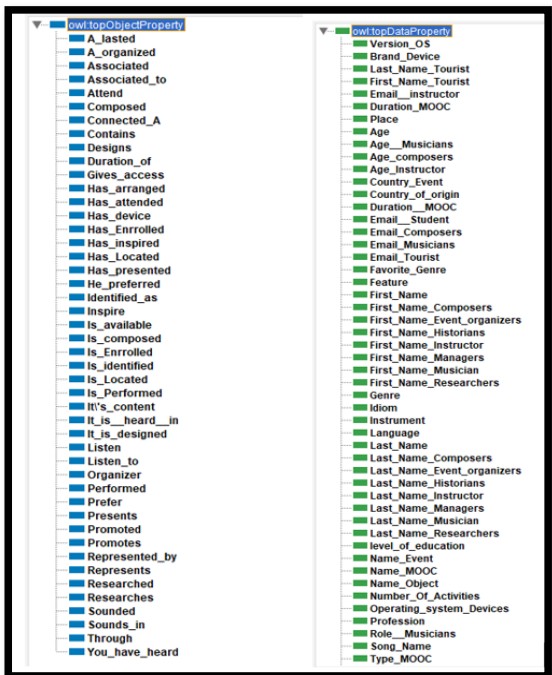

**Figure 4.** Object Property and Data Property. Obtained from Protégé software.

The object properties defined the relationships between the classes and subclasses of the model; the data properties corresponded to the instances of the ontology.

The relationships with their concepts and ranges are presented in Table 4.

**Table 4.** Ontology Relationships.

| Relationship | Concep | Range | Inverse |
|---|---|---|---|
| Connects_to | Devices | Net | Give_Acces |
| Contains | MOOC | Learning_Object | It's_content |
| Duration_of | Time | MOOC<br>Learning_Object<br>Cultural_Event | It_Last |
| Has_inspired | Vallenato_Songs | Learning_Object | Get_inspired |
| Identifies_as | Profile/Role | Users<br>Cultural_Managers<br>Instructors<br>Musicians<br>Tourists | It_is_identified_as |
| Is_available | Environment | Net | Has_arranged |
| Is_designed | MOOC | Cultural_Managers<br>Instructors | Has_designed |
| Is_heard_in | Devices | Vallenato_Songs | It_is_heard_by |
| Is_located | Location | Devices<br>Users<br>Environment<br>Cultural_Managers<br>Instructors<br>MOOC<br>Learning_Object<br>Net<br>Cultural_Event | Has_located |
| Is_presented_by | Cultural_Event | Composers<br>Musicians | Has_presented |
| Is_promoted_by | Cultural_Managers | Vallenato_Songs | Has_promoted |
| Is sung/Played by | Musicians | Vallenato_Songs | Has_sung/Played |
| Is written by | Composers | Vallenato_Songs | You_have_written |
| Is_Enrolled | User | MOOC | Has_Enrolled |
| It_is_through | Devices | Learning_Object<br>Cultural_Event | Is_through |
| Listen_to_me | Tourists | Vallenato_Songs | You've_heard |
| Organized_by | Cultural_Managers | Cultural_Event | Has_organized |
| Partners | Interface | MOOC<br>Learning_Object<br>Cultural_Event | Associated_to |
| Prefer | Preference | Users<br>Cultural_Managers<br>Instructors<br>Musicians<br>Tourists | Has_preferred |
| Presented | Vallenato_Songs | Cultural_Event | It_is_sung_in |
| Went | Tourists | Cultural_Event | Attend |
| You_have_accessed | Devices | Users<br>Cultural_Managers<br>Instructors<br>Musicians<br>Tourists | Access |

Using the Ontograf tab, Figure 5 shows the ontology with its respective relationships; each of the dotted lines of different colors represents the relationships between the concepts of the ontology.

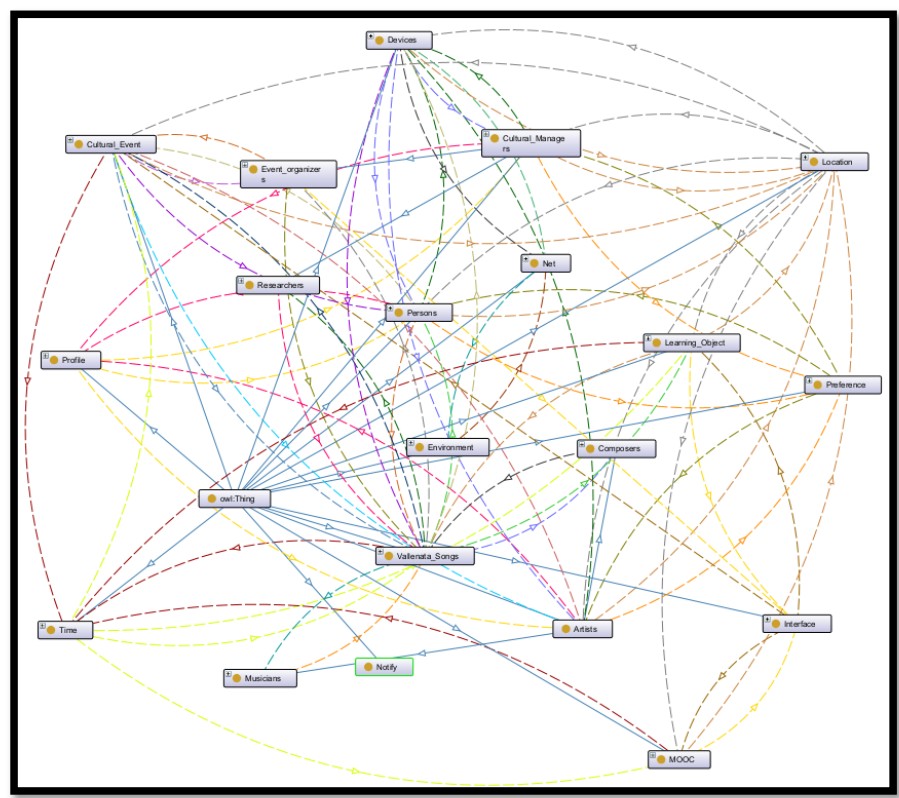

**Figure 5.** Ontology in Protégé. Obtained from Protégé software.

## 5. Situations or Behavior for a Vallenato Management Environment Supported by Contextual Awareness

The situations or behaviors are described below:

1.  A student of X age enrolls in a massive open, online course;
2.  Vallenata music artist A performs at a cultural event EC;
3.  Vallenata music composer C writes song V, which is performed at the CE event;
4.  The researcher designs a Massive Open Online Course inspired by the vallenata culture;
5.  Student A is enrolled in Massive Course 1 that was designed by a researcher, inspired by vallenata songs and contains a learning object 1;
6.  Student A receives a notification of enrollment in the Massive Open and Online Course;
7.  A Massive Open Online Course is inspired by one or more vallenato songs;
8.  The student has a learning object associated with his/her Massive Open Online Course;
9.  Instructor B designs a learning object inspired by a vallenata song composed by a composer C and performed by a musician M and inserts it in the Massive Open Online Course 1;
10. Tourist 1 attends the cultural event 1 where a vallenata music composer C and a musician M perform;
11. The system helps the tourist with the information of the vallenato musicians of his preference;
12. The system helps the students to choose the mass course of their preference taking into account the vallenato songs they listen to;
13. The system helps the tourist to locate the cultural event related to vallenato where the artists of his preference are performing;

14. The Event Organizer organizes an event related to vallenato that can be a festival, conversation, parranda, concert, among others;
15. The Manager represents the vallenato artists that are hired to perform at cultural events;
16. Students connect through their device to the Massive Open Online Course they are enrolled in;
17. Students solve the activities contained in the learning object of the Massive Open Online Course from their device;
18. People research about vallenato from their devices by reading NFC tags, QR codes located in different places of the Cultural Museum of Vallenato Music;
19. Composers show their unpublished songs from their devices;
20. People located in a U location review the location of cultural events related to vallenato;
21. Students connect via their device to the Massive Open Online Course they are enrolled in; the contents are adjusted to the network characteristics of the device.

## 6. Ontology Evaluation by Means of SWRL Rules

To evaluate the correct functioning of the ontology, some rules are established. This section shows nine situations or behaviors described in Section 5 are taken. The description includes each rule in its natural language and the SWRL, which was used because it allows expressing the OWL concepts in combination with RuleML [30].

In addition, a scenario of the situation posed is presented.

**Situation 1.** Student A receives notification of enrollment in the Massive Open Online Course.

**Rule 1.** We would like to list the students that are part of MOOC number 4.

**MOOC(?mooc), Has_Enrolled (MOOC004, ?std), Student(?std) -> Notify(?std)**

Figure 6 shows the results obtained from the situation presented, where the three notified students that are part of MOOC004 can be visualized.

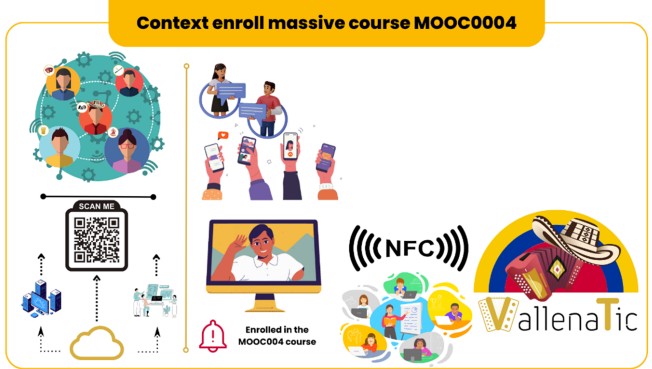

**Figure 6.** Diagram Situation 1.

As can be seen in Figure 7, students Std0002, Std0006, and Std0010 are enrolled in MOOC004.

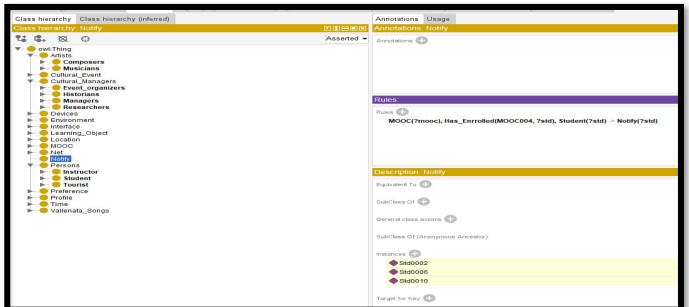

**Figure 7.** Result Rule 1. Obtained from Protégé software.

**Situation 2.** A student of age X enrolls in a massive open online course.
Figure 8 shows the diagram of the situation presented.

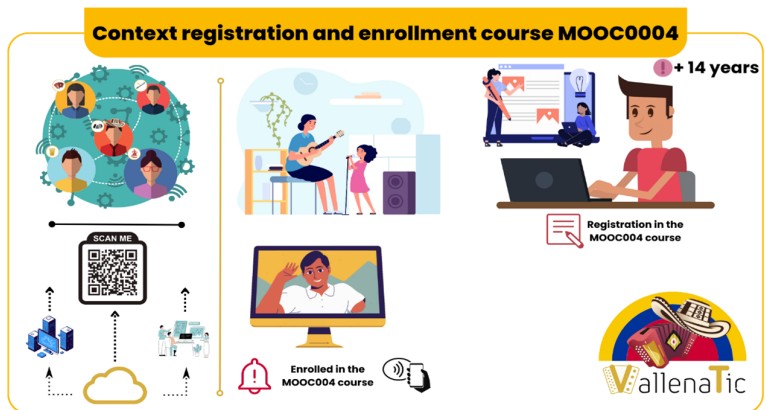

**Figure 8.** Diagram Situation 2.

**Rule 2:** List the students who are enrolled in the MOOC004 over 14 years of age.
**MOOC(?mooc), Is_Enrolled(?std, MOOC004), Student(?std), Age(?std, ?xage), greaterThan(?xage, 14) -> Notify(?std)**
The result is shown in Figure 9.

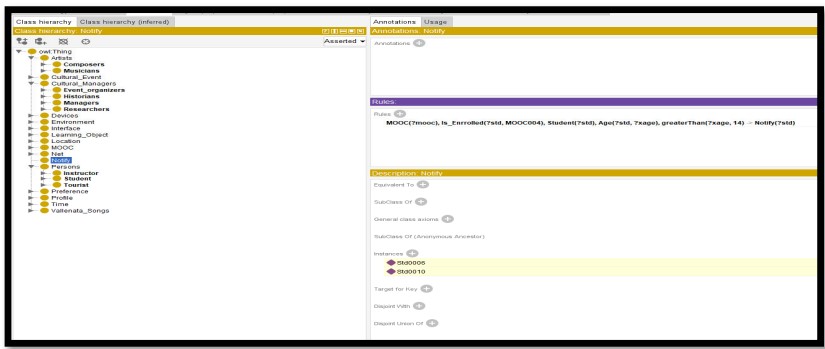

**Figure 9.** Result Rule 2. Obtained from Protégé software.

The Std0006 and Std0010 comply with Rule 2, are enrolled in MOOC0004, and are over 14 years of age.

**Situation 3.** Vallenata music artist A performs at an EC cultural event.
Figure 10 shows the diagram of the situation presented.

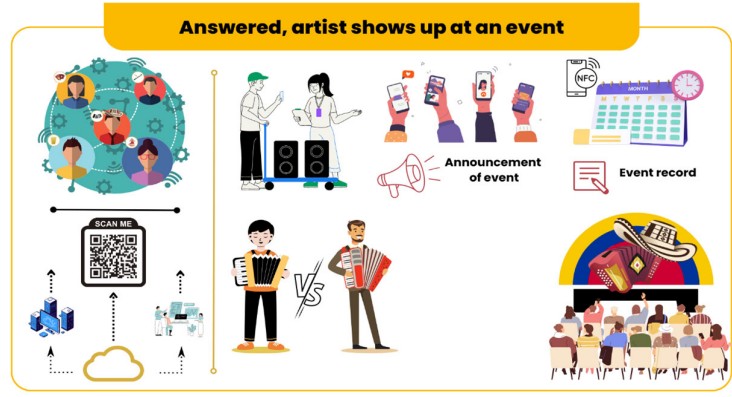

**Figure 10.** Diagram Situation 3.

**Rule 3.** List the vallenato artists performing at the Event0004 Cultural Event.
**Artists (?art), Cultural_Event(?event), Has_presented (?art, Event0004) -> Notify(?art)**
The result is shown in Figure 11.

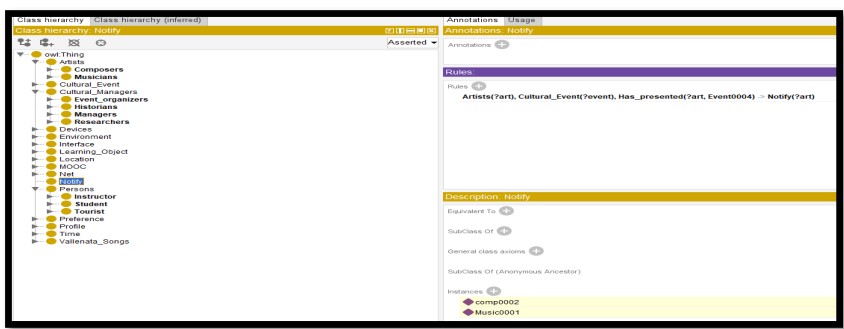

**Figure 11.** Result Rule 3. Obtained from Protégé software.

Figure 11 shows that the musician Music0001 and comp0002 are part of the presentations of the cultural event Event0004.

**Situation 4**. Tourist 1 attends cultural event 1 where a vallenata music composer C and a musician M perform.

Figure 12 shows the diagram of the situation presented.

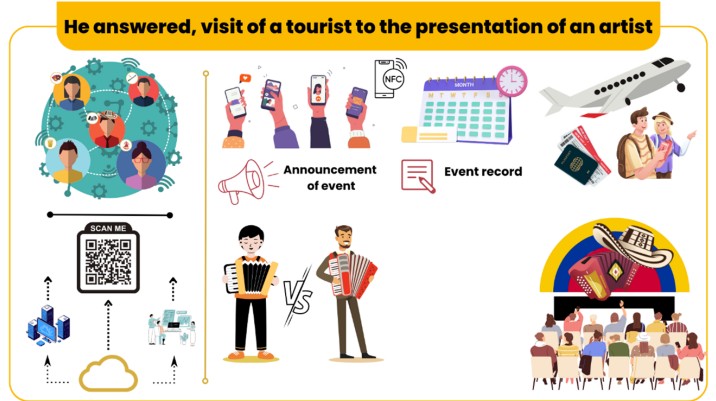

**Figure 12.** Diagram Situation 4.

**Rule 4.** Notify tourists and artists performing at the Event0003 Cultural Event.
**Tourist (?tou), Cultural_Event(?event), Attend (?tou, Event0003), Artists (?art), Cultural_Event(?event), Has_presented (?art, Event0003) -> Notify(?tou), Notify (?art)**
Figure 13 shows that Tourist Tourist0004 attends Event Event0003 where composer comp0001 and musician Music0002 perform.

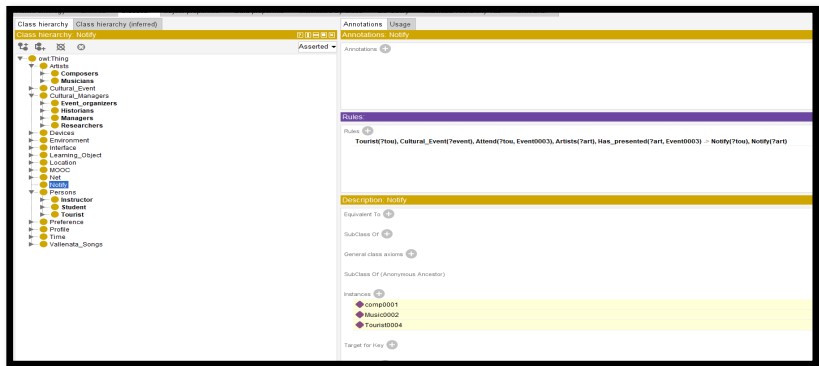

**Figure 13.** Result Rule 4. Obtained from Protégé software.

**Situation 5**. Student A is enrolled in Massive Course 1, which was designed by a researcher, inspired by vallenata songs, and contains a learning object 1.

Figure 14 shows the diagram of the situation presented.

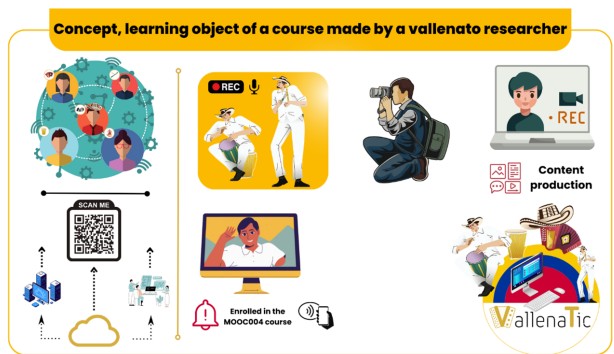

**Figure 14.** Diagram Situation 5.

**Rule 5.** List the students enrolled in MOOC004, the researcher who designed it, and the vallenato songs that inspired it.

**Vallenata_Songs(?vsng), MOOC(?mooc), Has_inspired (?vsng, MOOC004), MOOC (?mooc), Has_Enrrolled (MOOC004, ?std), Student(?std), Researchers (?res), MOOC (?mooc), Designs (?res, MOOC004), MOOC(?mooc), Learning_Object(?lob), It\\'s_content (?lob, MOOC004) -> Notify (?vsng), Notify(?std), Notify(?res), Notify(?lob)**

Figure 15 shows that students Std0002, Std0006, and Std0010 are enrolled in MOOC004, which was designed by the researcher Rese0004, who was inspired by the vallenato songs Sng0005, Sng0010, Sng0011, and Sng0012, which contains the Learning Object Object0004.

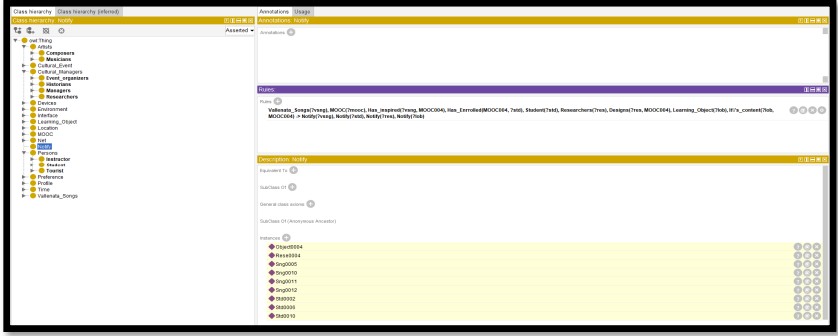

**Figure 15.** Result Rule 5. Obtained from Protégé software.

**Situation 6.** The Manager represents the vallenato artists who are hired to perform at cultural events and interpret vallenato songs.

Figure 16 shows the diagram of the situation presented.

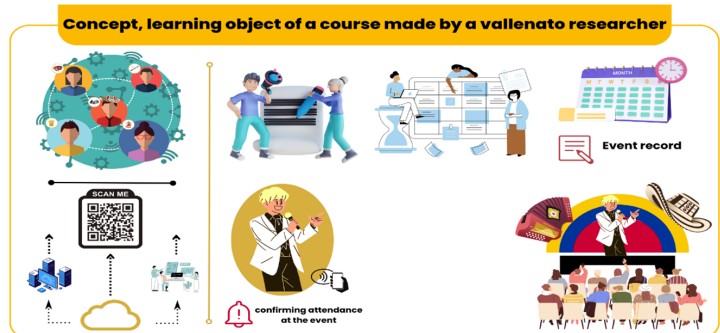

**Figure 16.** Diagram Situation 6.

**Rule 6.** List the representative of the musician Music0001, which establish the events where he performs and the songs he performs.

**Managers(?man), Musicians(?music), Represented_by (Music0001, ?man), Musicians(?music), Cultural_Event(?event), Has_presented (Music0001, ?event), Musicians (?music), Vallenata_Songs(?vsng), Performed (Music0001, ?vsng), -> Notify (?vsng), Notify(?man), Notify(?event)**

Figure 17 shows that Music Music0001 is represented by the representative Mang0002, he performs in the events Event0004 and Event0006 and interprets the vallenato songs Sng0004 and Sng0006.

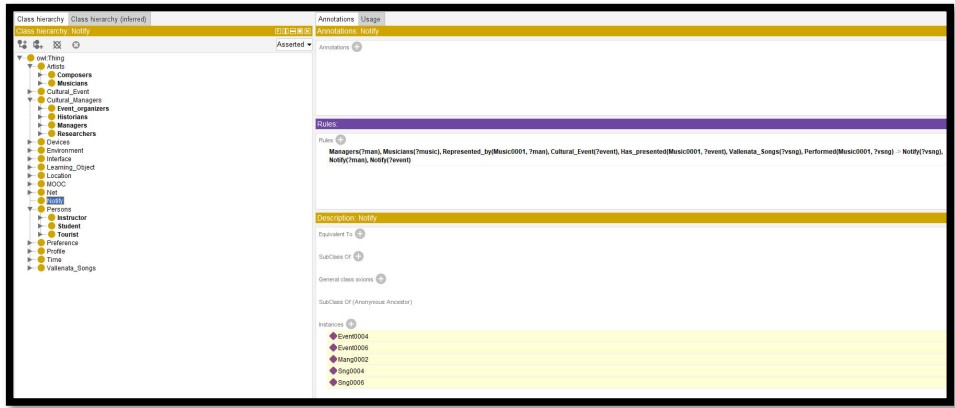

**Figure 17.** Result Rule 6. Obtained from Protégé software.

**Situation 7**: People located in a U location review the location of cultural events related to vallenato.

Figure 18 shows the diagram of the situation presented.

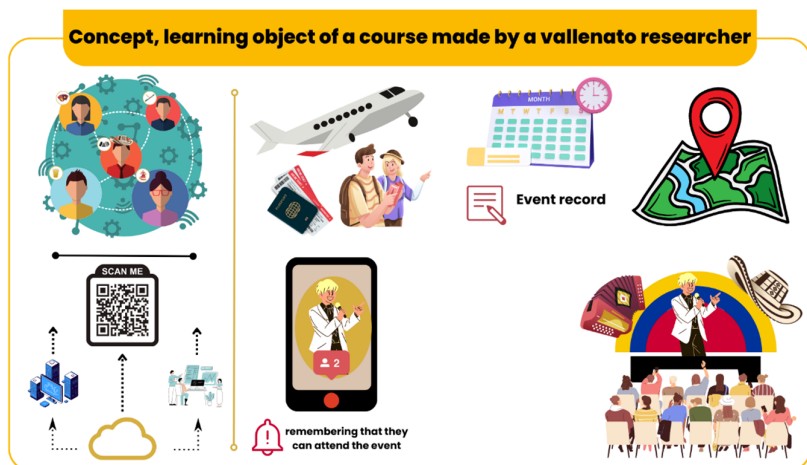

**Figure 18.** Diagram Situation 7.

**Rule 7:** Tourists in a LOC0001 Location review the location of cultural events in the same location.

**Tourist(?tou), Cultural_Event(?event), Location(?loc), Has_Located(LOC0001, ?tou), Has_Located(LOC0001, ?event) -> Notify(?event), Notify(?tou)**

As can be seen in Figure 19, tourists Tourist0008 and Tourist00012 who are at Location LOC0001 look for event information at their same location, which are Event0001 and Event0012.

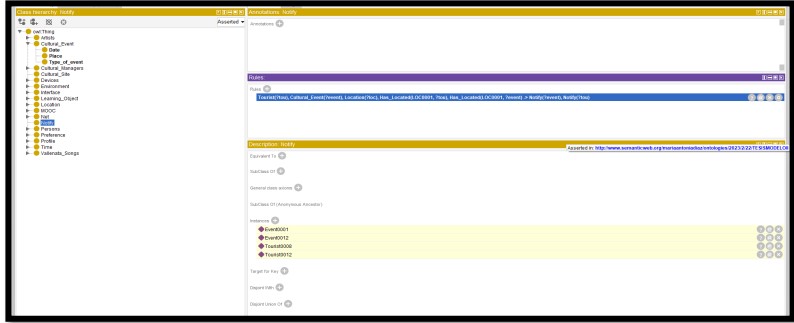

**Figure 19.** Result Rule 7. Obtained from Protégé software.

**Situation 8:** Students connect through their device to the Massive Open Online Course they are enrolled in; the contents are adjusted to the characteristics of the device's network.

Figure 20 shows the diagram of the situation presented.

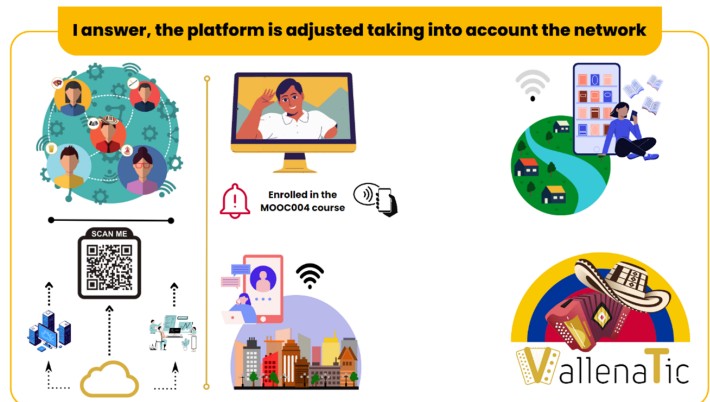

**Figure 20.** Diagram Situation 8.

**Rule 8:** Students enrolled in MOOC0004 connect to the course from their devices and are given the network access the two types of course content (Light or Robust)

**MOOC(?mooc), Has_Enrolled (MOOC004, ?std), Student(?std), Student(?std), Devices(?dev), Through (?dev, ?std), Devices(?dev), Net(?net), Connected_A (?dev, ?net), Net(?net), Type_of_content (?toc), Access_content (?net, ?toc) -> Notify(?std), Notify (?dev), Notify(?net), Notify (?toc)**

As seen in Figure 21, the notified students accessing MOOC0004 are: Std0002, Std0006, and Std0010, which use Devices Dev0001, Dev0003, and Dev0005, which connect to the network and adjust the contents considering the network that the devices have.

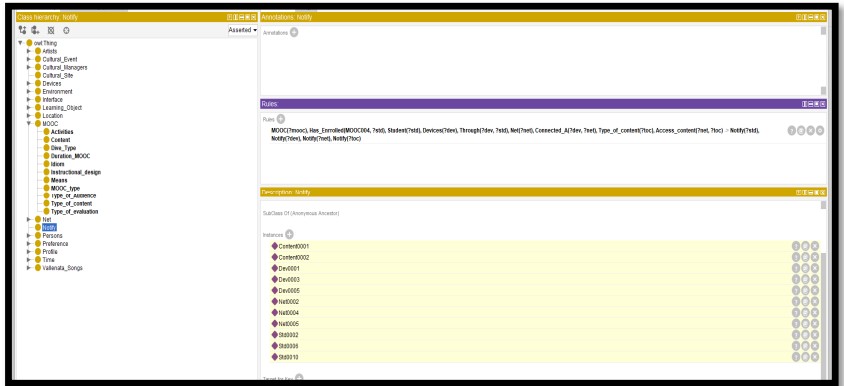

**Figure 21.** Result Rule 8. Obtained from Protégé software.

**Situation 9**. People research about vallenato from their devices by reading NFC tags, QR code located in different places of the Cultural Museum of Vallenato Music.

Figure 22 shows the diagram of the situation presented.

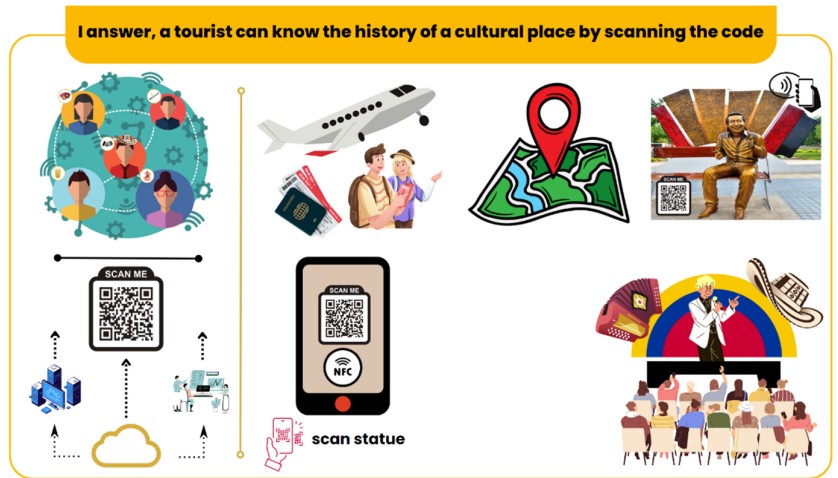

**Figure 22.** Diagram Situation 9.

**Rule 9:** A tourist/student visiting the cultural site Site0001 (Cultural Museum of vallenato music) has a device that allows him to research about vallenato since his device has NFC tags or QR code accessing a MOOC on the platform.

**MOOC(?moocnfc_QRcode), Student(?std), Has_Enrolled(?moocnfc_QRcode, ?std), Tourist(?tou), Cultural_Site(?site), Visited(Site0001, ?tou), Student(?std), Cultural_Site (?site), Visited(Site0001, ?std), Tourist(?tou), Devices(?devnfc_QRcode), Through(?devnfc_QRcode, ?tou), Student(?std), Devices(?devnfc_QRcode), Through (?devnfc_QRcode, ?std) -> Notify(?std), Notify(?tou), Notify(?moocnfc_QRcode), Notify (?devnfc_QRcode)**

Figure 23 shows the result of the rule after using the Pellet reasoner.

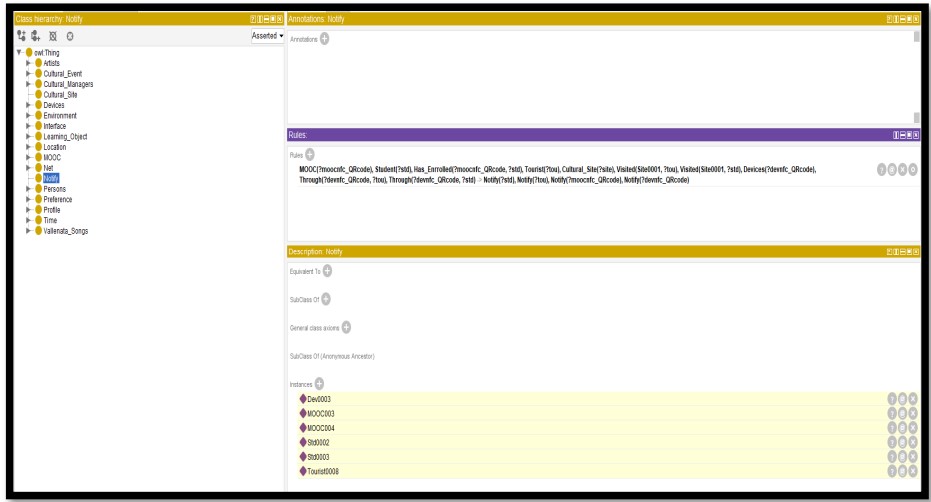

**Figure 23.** Result Rule 9. Obtained from Protégé software.

As shown in Figure 23, students Std0002 and Std0003 and Tourist Tourist0008 visit the cultural site Site0001 (Cultural Museum of Vallenata Music) and access information using their device by reading, which connect to the network and adjust the contents considering the network that the devices have.

## 7. Comparison of the Proposed Ontology Model with Other Ontologies

Considering that the modeling of ontologies has been widely studied, there is a wide range of ontologies with different purposes, but there is none like the one presented in this research; even so, a comparison is made with other ontology models designed for cultural heritage management.

The criteria used for the comparison of the classes in each context model are the following:

- Explicit definition of the class in the context model (✓);
- Explicit definition of the class by means of subclasses in the context model (✓/);
- Partial definition of the class through a single class or subclass contained in the model (P);
- Partial definition of the class through different classes or subclasses contained in the model (P/);
- Class not explicitly or partially defined (X).

Table 5 shows the comparison of the proposed ontology model "Vallenatic" with other ontologies for cultural heritage.

**Table 5.** Comparison of the proposed ontology model with other ontologies.

| Ontological Model | Purpose | Artist | Devices | Persons | Environment | Cultural Managers | Interface | Location | MOOC | Learning Object | Profile | Preference | Net | Time | Cultural Event | Vallenatas Songs | Cultural Site |
|---|---|---|---|---|---|---|---|---|---|---|---|---|---|---|---|---|---|
| **ONTOLOGIES FOR CULTURAL HERITAGE MANAGEMENT** | | | | | | | | | | | | | | | | | |
| ArCo (Arquitecture of Knowledge) [11] | Create a network of ontologies to represent cultural heritage data and publish the General Catalog proposed by the Italian Ministry of Culture. | X | X | P | ✓ | P | ✓ | ✓ | X | X | X | X | X | X | ✓ | X | ✓ |
| MOM [13] | It is an ontology for the management of cultural heritage, takes classes of the following Ontologies: CIDOC CRM, EDM, ORE, (https://www.openarchives.org/ore/1.0/datamodel) FOAF (http://www.foaf-project.org/), DC (https://www.dublincore.org/specifications/dublin-core/dces/) y SKOS (https://www.w3.org/2004/02/skos/, accessed on 3 February 2023). | X | X | ✓ | X | P/ | X | ✓ | X | ✓/ | X | X | X | ✓/ | ✓ | X | ✓ |
| CURIOCITY (Cultural Heritage for Urban Tourism in Indoor/Outdoor environments of the CITY) [12] | Designed to represent cultural heritage knowledge based on UNESCO definitions. | ✓ | P/ | ✓ | P/ | X | X | ✓ | X | X | ✓/ | X | X | ✓/ | P/ | P/ | ✓ |
| CIDOC CRM [7]. | It is the largest and most complex ontology in terms of cultural heritage, has 99 classes and 188 properties. | P/ | X | ✓ | ✓/ | X | P/ | P/ | X | X | P/ | P/ | X | P/ | ✓/ | X | ✓ |

**Table 5.** *Cont.*

| Ontological Model | Purpose | Artist | Devices | Persons | Environment | Cultural Managers | Interface | Location | MOOC | Learning Object | Profile | Preference | Net | Time | Cultural Event | Vallenatas Songs | Cultural Site |
|---|---|---|---|---|---|---|---|---|---|---|---|---|---|---|---|---|---|
| | ONTOLOGIES FOR LEARNING ENVIRONMENTS | | | | | | | | | | | | | | | | |
| An Ontology-based Framework or Context-aware Adaptive E-learning System [15] | Categorization of contextual information and modeling, along with the use of ontology to explicitly specify the context of the learner in an e-learning environment. | X | ✓ | ✓/ | X | X | X | ✓/ | X | X | X | P/ | X | ✓/ | X | X | X |
| A simple model of smart learning environment [16] | An ontology model for intelligent learning environments supported by contextual awareness. | X | ✓ | ✓/ | X | X | X | ✓ | X | ✓ | ✓/ | ✓/ | X | ✓ | X | X | X |
| A model for learning objects adaptation in light of mobile and context-aware computing [17] | Architectural model for the adaptation of objects from Learning considering device characteristics, learning style, and contextual information from Other Students | X | ✓ | ✓/ | X | X | ✓ | ✓ | X | ✓ | ✓ | X | X | X | X | X | X |
| Vallenatic Proposed model | An ontological model for the representation of vallenato as cultural heritage in a context-aware | ✓ | ✓ | ✓ | ✓ | ✓ | ✓ | ✓ | ✓ | ✓ | ✓ | ✓ | ✓ | ✓ | ✓ | ✓ | ✓ |

As can be seen in Table 4, the systematic literature review did not find a model that contemplates the protection of musical cultural heritage mediated by educational processes as proposed in this research.

## 8. Conclusions

This document contains the ontological model proposed for the preservation of Colombian vallenato as intangible cultural heritage of humanity through educational processes with the implementation of massive open online courses (MOOC).

For the development of the ontological model the NeOn methodology was selected, taking as a reference the scenario I, the elements of the context that compose the ontology are the following: Artists, Device, People, Environment, Cultural Managers, Interface, Location, MOOC, Learning Object, Profile, Preference, Network, Time, Cultural Event, Vallenato Songs.

The relationships and each of the concepts that compose the ontological network are described as well as its implementation in the Protégé Software version 5.6.1.

The model was validated through the implementation of rules using the Protégé pellet reasoner, where it can be observed that the model works.

This model was compared with other models to establish similarities and differences, and it was possible to establish that there is no model equal to the one proposed, although there are some that share components (see Table 5).

## 9. Future Works

At the end of this document, recommendations for future research are made. The first one is to continue with the evaluation of the ontology model by implementing the other rules that were not contemplated in this document.

The second recommendation is focused on continuing with the development of the architecture that will consume the proposed ontology.

The third recommendation is to make use of the proposed model for other musical genres that, like vallenato, are recognized by UNESCO as intangible cultural heritage of humanity, such as Spanish flamenco, Argentine Tango, Mexican Mariachi music, Peruvian scissors dance, Brazilian capoeira, Dominican bachata, Jamaican reggae, among others.

**Author Contributions:** Introduction, M.A.D.-M.; Related Works, E.D.-L.-H.-F. and R.R.-V.; validation, M.A.D.-M.; Ontological Model, M.A.D.-M. and J.E.G.G.; Ontology evaluation, M.A.D.-M., R.R.-V., E.D.-L.-H.-F. and J.E.G.G.; ontology in Protégé, M.A.D.-M.; data curation, M.A.D.-M. and E.D.-L.-H.-F.; writing—original draft preparation, E.D.-L.-H.-F.; writing—review and editing, J.E.G.G. and M.A.D.-M.; visualization, M.A.D.-M.; supervision, E.D.-L.-H.-F. and R.R.-V.; project administration, E.D.-L.-H.-F. All authors have read and agreed to the published version of the manuscript.

**Funding:** This research is supported by the Ministry of Science and Technology of the Republic of Colombia through the Bicentenary Scholarship program in its first cohort. The authors did not receive support from any organization for the submitted work. No funding was received to assist with the preparation of this manuscript. No funding was received for conducting this study. No funds, grants, or other support was received.

**Data Availability Statement:** Not applicable.

**Acknowledgments:** This research is supported by the Ministry of Science and Technology of Colombia and the Universidad de la Costa CUC through the bicentennial scholarship program in its first cohort.

**Conflicts of Interest:** The authors declare no conflict of interest.

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
