# Peer review of "An Ontological Model for the Representation of Vallenato as Cultural Heritage in a Context-Aware System"

_heritage, doi:10.3390/heritage6080297_

Round 1
Reviewer 1 Report
According to the authors, the manuscript proposes an ontological model of vallenato as cultural heritage and presents the usage of this model in a context-aware system. The ontology is designed by means of an acknowledged methodology (NeOn methodology), which offers support to conduct the development process in a systematic way. Also, the proposed ontology is compared to related works in order to point out its contribution. In my opinion, it is an interesting work. Anyway, I would like to highlight some aspects that could improve the manuscript, as follows:
1) In the first paragraph of page 3, we have: “This paper presents a proposal for a MOOC-based contextual awareness system for the management of traditional Colombian vallenato”. But in the abstract, we have “this document proposes an ontological model for the representation of vallenato as cultural heritage in a context-aware system called Vallenatic.”. In my opinion, the abstract is more aligned with what the manuscript presents. The focus of the paper is on ontology (that is used in a context-aware system) and not directly on the system.
2) In the first paragraph of Section 2 (related work), the authors point out that a systematic literature review (SLR) was conducted, however, the results of the SLR seem to be not used in the paper. It is said that 204 articles were analyzed, but “so what”? The result of this analysis was not presented in the paper. It seems that the SLR was not relevant or required for designing the proposed ontology. In fact, about 10 works were discussed in this section, but what about the other papers identified by the SLR. It suggests that if the authors remove the first paragraph from the manuscript, it would not be missed.
3) The authors used the term “domain” to refer to elements that seem to be concepts of the ontology. I did not understand why to use “domain” and not “concept”. The notion of domain uses to refer to a subject (vallenato) about what an ontology aims at describing; a domain, in turn, is characterized by a number of concepts, relations, and axioms.
4) About the functional requirements: why were not the competence questions presented? I missed them.
5) In “Task 8 : Extraction of terminology and its frequency”, the authors have presented a word cloud (Figure 1), however, It is not clear how it is useful. How do the information offered by the word cloud used in the ontology design? The authors said: “As can be seen in Figure 1, the words with the highest frequencies are those related to the genres of vallenato songs (merengue, son, paseo, and puya) followed by singers, inspiration, lyrics, nature, among others”. But… so what?
6) In Section 3.3 a conceptual model is presented with concepts and their relations. However, the authors do not explain or define each concept. In my opinion, it lacks the definition of each concept and how it related to each other and also existing constraints. For example: what does “interface” mean? The notion of “interface” is different, depending on the application domain.
7) Why were not the subclasses (e.g., composers, musicians, managers, etc) included in the ontological model of Figure 2?
8) In Figure 3, the right side figure is difficult to see.
9) Figure 5 is too difficult to see/understand. In fact, I think it does not bring any contribution to the paper.
10) In Section 5 there are a number of said situations/behaviors. What is the relation between these situations/behaviors and the competence questions?
11) In my opinion, figures 6, 8, 10, 12, 14, 16, 18, 20, and 22 are not useful for understanding each scenario. They present a number of graphical elements that are not easily connected. Also, no excerpt is used to explain/introduce the figures.
12) In future works, I would like to suggest that other techniques were used to validate the ontology — for example, validation with specialists.
Author Response
Respectful greetings,
We appreciate the comments corresponding to the revision of our article, which has allowed us to increase the quality of the document and adapt it to the requirements of the journal. The document has change markers enabled and all modifications have been highlighted in red. Our response to your comments is presented below:
Point 1: In the first paragraph of page 3, we have: “This paper presents a proposal for a MOOC-based contextual awareness system for the management of traditional Colombian vallenato”. But in the abstract, we have “this document proposes an ontological model for the representation of vallenato as cultural heritage in a context-aware system called Vallenatic.”. In my opinion, the abstract is more aligned with what the manuscript presents. The focus of the paper is on ontology (that is used in a context-aware system) and not directly on the system.
Response 1: Thank you very much for your appreciation,
The term has been corrected to align it with what is presented in the manuscript.
Point 2: In the first paragraph of Section 2 (related work), the authors point out that a systematic literature review (SLR) was conducted, however, the results of the SLR seem to be not used in the paper. It is said that 204 articles were analyzed, but “so what”? The result of this analysis was not presented in the paper. It seems that the SLR was not relevant or required for designing the proposed ontology. In fact, about 10 works were discussed in this section, but what about the other papers identified by the SLR. It suggests that if the authors remove the first paragraph from the manuscript, it would not be missed.
Response 2: A paragraph was added stating the use of the result of the review, specifically stating that of all the articles reviewed, there is no ontology equal to the one presented in this manuscript.
Point 3: The authors used the term “domain” to refer to elements that seem to be concepts of the ontology. I did not understand why to use “domain” and not “concept”. The notion of domain uses to refer to a subject (vallenato) about what an ontology aims at describing; a domain, in turn, is characterized by a number of concepts, relations, and axioms.
Response 3: The term was changed to the one suggested by you.
Point 4: About the functional requirements: why were not the competence questions presented? I missed them.
Response 4: In consensus, all authors considered not placing the questions in the manuscript.
Point 5: In “Task 8 : Extraction of terminology and its frequency”, the authors have presented a word cloud (Figure 1), however, It is not clear how it is useful. How do the information offered by the word cloud used in the ontology design? The authors said: “As can be seen in Figure 1, the words with the highest frequencies are those related to the genres of vallenato songs (merengue, son, paseo, and puya) followed by singers, inspiration, lyrics, nature, among others”. But… so what?
Response 5: A paragraph is added explaining the usefulness of term extraction.
Point 6: In Section 3.3 a conceptual model is presented with concepts and their relations. However, the authors do not explain or define each concept. In my opinion, it lacks the definition of each concept and how it related to each other and also existing constraints. For example: what does “interface” mean? The notion of “interface” is different, depending on the application domain.
Response 6: Added the concepts
Point 7: Why were not the subclasses (e.g., composers, musicians, managers, etc) included in the ontological model of Figure 2?
Response 7: The image was enlarged and difficult to see.
Point 8: In Figure 3, the right side figure is difficult to see.
Response 8: Image enlarged for better visualization
Point 9: Figure 5 is too difficult to see/understand. In fact, I think it does not bring any contribution to the paper.
Response 9: A paragraph is added explaining the relationships.
Point 10: In Section 5 there are a number of said situations/behaviors. What is the relation between these situations/behaviors and the competence questions?
Response 10: These situations allow the evaluation of the Ontology and are the ones to be solved by the finished system.
Point 11: In my opinion, figures 6, 8, 10, 12, 14, 16, 18, 20, and 22 are not useful for understanding each scenario. They present a number of graphical elements that are not easily connected. Also, no excerpt is used to explain/introduce the figures.
Response 11: A paragraph explaining the content of the figures was added.
Point 12: In future works, I would like to suggest that other techniques were used to validate the ontology — for example, validation with specialists.
Response 12: Thank you very much for your feedback and we will take it into account for future work.
We thank you again for your recommendations.

Reviewer 2 Report
The paper proposes an ontological model for representing vallenato as cultural heritage in a context-aware system called Vallenatic. The proposed model can be used for other musical genres that have the recognition of cultural and intangible heritage.
The paper is well written. However, a considerable disadvantage. It does not provide a reference to the ontologies and rules it describes.
Minor comments:
1. Figure 1 seems to me that is not needed.
2. Figure 2 can be more readable.
3. In Table 3 the are a lot of inconsistencies. Two-word ontology elements sometimes have whitespace and sometimes are separated by _. The _ sign seems to be better.
Proofreading is needed.
Author Response
Respectful greetings,
We appreciate the comments corresponding to the revision of our article, which has allowed us to increase the quality of the document and adapt it to the requirements of the journal. The document has change markers enabled and all modifications have been highlighted in red. Our response to your comments is presented below:
Point 1: Figure 1 seems to me that is not needed.
Response 1: Thank you very much for your appreciation,
A paragraph is added explaining the usefulness of term extraction.
Point 2: Figure 2 can be more readable.
Response 2: Image enlarged for better visualization
Point 3: In Table 3 the are a lot of inconsistencies. Two-word ontology elements sometimes have whitespace and sometimes are separated by _. The _ sign seems to be better.
Response 3: Corrected, it should be specified that it is Table 4.
We thank you again for your recommendations.
